# Nearly Tight Black-Box Auditing of Differentially Private Machine Learning

**Meenatchi Sundaram Muthu Selva Annamalai**
University College London
meenatchi.annamalai.22@ucl.ac.uk

**Emiliano De Cristofaro**
University of California, Riverside
emilianodc@cs.ucr.edu

## Abstract

This paper presents an auditing procedure for the Differentially Private Stochastic Gradient Descent (DP-SGD) algorithm in the black-box threat model that is substantially tighter than prior work. The main intuition is to craft worst-case initial model parameters, as DP-SGD's privacy analysis is agnostic to the choice of the initial model parameters. For models trained on MNIST and CIFAR-10 at theoretical $\varepsilon = 10.0$, our auditing procedure yields empirical estimates of $\varepsilon_{emp} = 7.21$ and $6.95$, respectively, on a 1,000-record sample and $\varepsilon_{emp} = 6.48$ and $4.96$ on the full datasets. By contrast, previous audits were only (relatively) tight in stronger white-box models, where the adversary can access the model's inner parameters and insert arbitrary gradients. Overall, our auditing procedure can offer valuable insight into how the privacy analysis of DP-SGD could be improved and detect bugs and DP violations in real-world implementations. The source code needed to reproduce our experiments is available from https://github.com/spalabucr/bb-audit-dpsgd.

## 1   Introduction

Differentially Private Stochastic Gradient Descent (DP-SGD) [1] allows training machine learning models while providing formal Differential Privacy (DP) guarantees [15]. DP-SGD is a popular algorithm supported in many open-source libraries like Opacus [40], TensorFlow [19], and JAX [4]. Progress in privacy accounting and careful hyper-parameter tuning has helped reduce the gap in model performance compared to non-private training [11]. Furthermore, pre-training models on large public datasets and then fine-tuning them on private datasets using DP-SGD can also be used to improve utility [11].

Implementing DP-SGD correctly can be challenging – as with most DP algorithms [5, 26] – and bugs that lead to DP violations have been found in several DP-SGD implementations [7, 22]. These violations do not only break the formal DP guarantees but can also result in realistic privacy threats [17, 33]. This prompts the need to audit DP-SGD implementations to verify whether the *theoretical* DP guarantees hold in practice. This usually involves running membership inference attacks [32], whereby an adversary infers whether or not a sample was used to train the model. The adversary's success estimates the *empirical* privacy leakage from DP-SGD, and that is compared to the theoretical DP bound. If the former is *"far"* from the latter, it means that the auditing procedure is *loose* and may not be exploiting the maximum possible privacy leakage. Whereas, when empirical and theoretical estimates are *"close,"* auditing is *tight* and can be used to identify bugs and violations [28].

38th Conference on Neural Information Processing Systems (NeurIPS 2024).

Although techniques to tightly audit DP-SGD exist in literature [28, 29], they do so only in *active white-box* threat models, where the adversary can observe and insert arbitrary gradients into the intermediate DP-SGD steps. By contrast, in the more restrictive *black-box* threat model, where an adversary can only insert an input canary and observe the final trained model, prior work has only achieved loose audits [20, 28, 29]. In fact, prior work has suggested that the privacy analysis of DP-SGD in the black-box threat model can be tightened but only in limited settings (i.e., strongly convex smooth loss functions) [3, 9, 39]. Therefore, it has so far remained an open research question [28] whether or not it is possible to tightly audit DP-SGD in the black-box model.

In this paper, we introduce a novel auditing procedure that achieves substantially tighter black-box audits of DP-SGD than prior work, even for *natural* (i.e., not adversarially crafted) datasets. The main intuition behind our auditing procedure is to craft worst-case *initial model parameters*, as DP-SGD's privacy analysis is agnostic to the choice of initial model parameters. We show that the gap between worst-case empirical and theoretical privacy leakage in the black-box threat model is much smaller than previously thought [20, 28, 29]. When auditing a convolutional neural network trained on MNIST and CIFAR-10 at theoretical $\varepsilon = 10$, our audits achieve an empirical privacy leakage estimate of $\varepsilon_{emp} = 6.48$ and $4.96$, respectively, compared to $\varepsilon_{emp} = 3.41$ and $0.69$ when the models were initialized to average-case initial model parameters, as done in prior work [11, 20, 28].

In the process, we identify and analyze the key factors affecting the tightness of black-box auditing: 1) dataset size and 2) gradient clipping norm. Specifically, we find that the audits are looser for larger datasets and larger gradient clipping norms. We believe this is due to more noise, which reduces the empirical privacy leakage. When considering smaller datasets with 1,000 samples, our audits are tighter – for $\varepsilon = 10.0$, we obtain $\varepsilon_{emp} = 7.21$ and $6.95$, respectively, for MNIST and CIFAR-10.

Finally, we present tight audits for models with only the last layer fine-tuned using DP-SGD. Specifically, we audit a 28-layer Wide-ResNet model pre-trained on ImageNet-32 [11] with the last layer privately fine-tuned on CIFAR-10. With $\varepsilon = 10.0$, we achieve an empirical privacy leakage estimate of $\varepsilon_{emp} = 8.30$ compared to $\varepsilon_{emp} = 7.69$ when the models are initialized to average-case initial parameters (as in previous work [11, 20, 28]).

Overall, the ability to perform rigorous audits of differentially private machine learning techniques allows shedding light on the tightness of theoretical guarantees in different settings. At the same time, it provides critical diagnostic tools to verify the correctness of implementations and libraries.

## 2 Related Work

**DP-SGD.** Abadi et al. [1] introduce DP-SGD and use it to train a feed-forward neural network model classifying digits on the MNIST dataset. On the more complex CIFAR-10 dataset, they instead fine-tune a pre-trained convolutional neural network, incurring a significant performance loss. Tramèr and Boneh [35] improve on [1] using hand-crafted feature extraction techniques but also suggest that end-to-end differentially private training may be inherently difficult without additional data. Papernot et al. [30] and Dörmann et al. [14] also present improvements considering different activation functions and larger batch sizes. Finally, De et al. [11] achieve state-of-the-art model performance on CIFAR-10 both while training from scratch and in fine-tuning regimes; they do so by using large batch sizes, augmenting training data, and parameter averaging. Our work focuses on auditing realistic models with high utility that might be used in practice; thus, we audit models with utility similar to [11] and the same models pre-trained in [11] in the fine-tuning regime.

**Membership Inference Attacks (MIAs).** Shokri et al. [32] propose the first black-box MIAs against machine learning models using shadow models. Carlini et al. [6] improve by fitting a Gaussian distribution on the loss of the target sample and using fewer models to achieve equivalent attack accuracies. Sablayrolles et al. [31] consider per-example thresholds to improve the performance of the attacks in a low false-positive rate setting [6]. By factoring in various uncertainties in the MIA "game," Ye et al. [38] construct an optimal generic attack. Concurrent work [16, 25, 37] use poisoned pre-trained models, which is somewhat similar to our approach of using worst-case initial model parameters, but with a different focus. Specifically, Liu et al. [25] and Wen et al. [37] show that privacy leakage from MIAs is amplified when pre-trained models are poisoned but, crucially, do not consider DP auditing. Additionally, while Feng and Tramèr [16] achieve tight audits with poisoned pre-trained models, their strategy is geared towards the last-layer only fine-tuning regime. By contrast,

we focus on auditing DP-SGD with full model training to estimate the empirical worst-case privacy leakage in the black-box setting.

**Auditing DP-SGD.** Jayaraman and Evans [21] derive empirical privacy guarantees from DP-SGD, observing a large gap from the theoretical upper bounds. Jagielski et al. [20] use data poisoning in the black-box model to audit Logistic Regression and Fully Connected Neural Network models and produce tighter audits than using MIAs. They initialize models to fixed (average-case) initial parameters, but the resulting empirical privacy guarantees are still far from the theoretical bounds. They also adapt their procedure to the fine-tuning regime, again producing loose empirical estimates.

Nasr et al. [29] are the first to audit DP-SGD tightly; they do so by using adversarially crafted datasets and *active white-box* adversaries that insert canary gradients into the intermediate steps of DP-SGD. In follow-up work, [28] presents a tight auditing procedure also for natural datasets, again in the *active white-box* model. Zanella-Béguelin et al. [42] present tighter auditing procedures using Bayesian techniques, needing fewer models to audit larger privacy levels. Finally, De et al. [11] perform black-box auditing of their implementation of DP-SGD in the JAX framework [4]; they also use fixed (average-case) initialization of models as in [20], similarly achieving loose audits.

Overall, while prior work [11, 20, 29] showed that reducing the randomness in the initial model parameters results in tighter empirical privacy leakage estimates, the audits remained loose as the fixed model parameters were still initialized *randomly*. Specifically, designing an effective adversarial strategy that provides tight audits in the black-box setting without destroying utility has remained, to the best of our knowledge, an open problem.

## 3 Background

### 3.1 Differential Privacy (DP)

**Definition 1** (Differential Privacy (DP) [15])**.** A randomized mechanism $\mathcal{M} : \mathcal{D} \rightarrow \mathcal{R}$ is $(\varepsilon, \delta)$-differentially private if, for any two neighboring datasets $D, D' \in \mathcal{D}$ and $S \subseteq \mathcal{R}$, it holds:

$$\Pr[\mathcal{M}(D) \in S] \leq e^{\varepsilon} \Pr[\mathcal{M}(D') \in S] + \delta$$

Put simply, DP guarantees a formal upper bound (constrained by the privacy parameter $\varepsilon$) on the probability that any adversary observing the output of the mechanism $\mathcal{M}$ can distinguish between two *neighboring* inputs to $\mathcal{M}$ – i.e., two datasets differing in only one record.

### 3.2 Differentially Private Machine Learning (DPML)

**DP-SGD.** Differentially Private Stochastic Gradient Descent (DP-SGD) [1] is a popular algorithm for training machine learning models while satisfying DP. In this paper, we consider classification tasks on samples from the domain $\mathcal{X} \times \mathcal{Y}$, where $\mathcal{X}$ is the features domain and $\mathcal{Y}$ is the labels domain.

DP-SGD, reviewed in Algorithm 1, introduces two hyper-parameters: gradient clipping norm $C$ and noise multiplier $\sigma$. Typically, $C$ is set to 1, while $\sigma$ is calibrated to the required privacy level $(\varepsilon, \delta)$ based on the number of iterations, $T$, and batch size, $B$. Early DP-SGD work [1] set batch sizes to be small and similar to those of non-private SGD; however, recent work has highlighted the advantage of setting large batch sizes [11, 14] by training state-of-the-art models with them. Following this trend, and to ease auditing, we set $B = N$, as also done in prior work [28]. Finally, while the initial set of model parameters $\theta_0$ is typically sampled randomly (e.g., using Xavier initialization [18]), it can also be set arbitrarily (and even adversarially), without affecting the privacy guarantees provided by DP-SGD.

**Fine-tuning regime.** In this setting, one first pre-trains a model on a (large) public dataset and then fine-tunes it on the private data. When a suitable public dataset is available, prior work has shown that fine-tuning can achieve much higher utility than training from scratch using DP-SGD [11].

There are two methods for fine-tuning: doing it for all layers or only the last one. In both cases, the initial model parameters, $\theta_0$, are set to the pre-trained model parameters (rather than randomly) *before* running DP-SGD. However, while the former updates all model parameters during DP-SGD, with the latter, the model parameters until the last (fully connected) layer are "frozen" and not updated during DP-SGD. This setting treats the pre-trained model as a feature extractor and trains a Logistic

---

**Algorithm 1** Differentially Private Stochastic Gradient Descent (DP-SGD) [1]

---

**Require:** Dataset, $D$. Iterations, $T$. Learning rate, $\eta$. Batch size, $B$. Loss function, $\ell$. Initial model parameters, $\theta_0$. Noise multiplier, $\sigma$. Clipping norm, $C$.

1: **for** $t \in [T]$ **do**
2:      Sample $L_t$ from $D$ with sampling probability $B/N$
3:      **for** $(x_i, y_i) \in L_t$ **do**
4:         $g_t(x_i, y_i) \leftarrow \nabla_{\theta_t} \ell(\theta_t; (x_i, y_i))$
5:         $\bar{g}_t(x_i, y_i) \leftarrow g_t(x_i) / \max(1, \frac{||g_t(x_i)||_2}{C})$
6:      **end for**
7:      $\tilde{g}_t \leftarrow \frac{1}{B} \left( \sum_i \bar{g}_t(x_i, y_i) + \mathcal{N}(0, C^2 \sigma^2 \mathbb{I}) \right)$
8:      $\theta_{t+1} \leftarrow \theta_t - \eta \tilde{g}_t$
9: **end for**
10: **return** $\theta_T$

---

Regression model using DP-SGD based on the extracted features. As prior work shows [11] that this setting produces models with accuracy closer to the non-private state of the art (while also incurring smaller computational costs), we also do fine-tuning of just the last layer.

### 3.3 DP Auditing

As mentioned above, DP provides a theoretical limit on the adversary's ability – bounded by the privacy parameter $\varepsilon$ – to distinguish between $\mathcal{M}(D)$ and $\mathcal{M}(D')$. With auditing, one assesses this limit empirically to verify whether the theoretical guarantees provided by $\mathcal{M}$ are also met in practice. There may be several reasons this does not happen, e.g., the privacy analysis of $\mathcal{M}$ is not tight [29] or because of bugs in the implementation of $\mathcal{M}$ [28, 36]. Therefore, with DP mechanisms increasingly deployed in real-world settings, auditing them becomes increasingly important.

**Deriving $\varepsilon_{emp}$.** When auditing a DP mechanism $\mathcal{M}$, one runs $\mathcal{M}$ repeatedly on two neighboring inputs, $D$ and $D'$, $R$ times each and the outputs ($O = \{o_1, ..., o_R\}$ and $O' = \{o'_1, ..., o'_R\}$) are given to an adversary $\mathcal{A}$. Next, $\mathcal{A}$ tries to distinguish between $O$ and $O'$, which determines a false positive rate (FPR) and false negative rate (FNR). For a given number of outputs $2R$ and confidence level $\alpha$, empirical upper bounds $\overline{\text{FPR}}$ and $\overline{\text{FNR}}$ can be computed using Clopper-Pearson confidence intervals [29, 28]. Finally, the empirical upper bounds are converted to an empirical lower bound $\varepsilon_{emp}$ using the *privacy region* of $\mathcal{M}$ (see below) and $\varepsilon_{emp}$ can be compared to the theoretical $\varepsilon$. We consider an audit *tight* if $\varepsilon_{emp} \approx \varepsilon$. To ease presentation, we abstract away the details and use $\varepsilon_{emp} \leftarrow \text{EstimateEps}(\text{FPR}, \text{FNR}, 2R, \alpha, \delta)$ to denote the process of estimating an $\varepsilon_{emp}$ from a given FPR and FNR.[1]

**Privacy region of DP-SGD.** Given a mechanism $\mathcal{M}$, the privacy region of the mechanism, $\mathcal{R}_{\mathcal{M}}(\varepsilon, \delta) = \{(\text{FPR}, \text{FNR}) | \cdot \}$ defines the FPR and FNR values attainable for any adversary aiming to distinguish between $\mathcal{M}(D)$ and $\mathcal{M}(D')$. Nasr et al. [28] note that the privacy region of DP-SGD corresponds (tightly) to that of $\mu$-Gaussian Differential Privacy [13], which can be used to first compute a lower bound on $\mu$:

$$\mu_{emp} = \Phi^{-1}(1 - \overline{\text{FPR}}) - \Phi^{-1}(\overline{\text{FNR}}) \tag{1}$$

Then, a lower bound on $\mu$ corresponds to a lower bound on $\varepsilon$ by means of the following theorem:

**Theorem 1** ($\mu$-GDP to $(\varepsilon, \delta)$-DP conversion [13]). A mechanism is $\mu$-GDP iff it is $(\varepsilon, \delta(\varepsilon))$-DP for all $\varepsilon \geq 0$, where:

$$\delta(\varepsilon) = \Phi\left(-\frac{\varepsilon}{\mu} + \frac{\mu}{2}\right) - e^{\varepsilon} \Phi\left(-\frac{\varepsilon}{\mu} - \frac{\mu}{2}\right) \tag{2}$$

## 4 Auditing Procedure

In this section, we present our auditing procedure. We discuss the threat model in which we operate, the key intuition for using *worst-case initial parameters*, and how we craft them to maximize the empirical privacy leakage.

---

[1]Please refer to `https://github.com/spalabucr/bb-audit-dpsgd` for details.

---

**Algorithm 2** Auditing DP-SGD

---

**Require:** Data distribution, $\mathcal{D}$. Number of samples, $n$. Initial model parameters, $\theta_0$. Loss function, $\ell$. Number of repetitions, $2R$. Decision threshold, $\tau$. Confidence level, $\alpha$.

1: Sample $D \sim \mathcal{D}^{n-1}$
2: Craft target sample $(x_T, y_T)$
3: $D' = D \cup \{(x_T, y_T)\}$

4: Observations $O \leftarrow \{\}, O' \leftarrow \{\}$
5: **for** $r \in [R]$ **do**
6:     $\theta_r \leftarrow \text{DP-SGD}(D; \theta_0, \ell)$
7:     $\theta'_r \leftarrow \text{DP-SGD}(D'; \theta_0, \ell)$
8:     $O[t] \leftarrow \ell(\theta_r; (x_T, y_T))$
9:     $O'[t] \leftarrow \ell(\theta'_r; (x_T, y_T))$
10: **end for**

11: $\text{FPR} \leftarrow \frac{1}{R}|\{o|o \in O, o \geq \tau\}|$
12: $\text{FNR} \leftarrow \frac{1}{R}|\{o|o \in O', o < \tau\}|$
13: $\varepsilon_{emp} \leftarrow \text{EstimateEps}(\text{FPR}, \text{FNR}, 2R, \alpha, \delta)$
14: **return** $\varepsilon_{emp}$

---

**Threat model.** We consider a simple *black-box* model where the adversary only sees the final model parameters, $\theta_T$, from DP-SGD. Specifically, unlike prior work on tight audits [28, 29], the adversary cannot observe intermediate model parameters or insert canary gradients. We do so to consider more realistic adversaries that may exist in practice. Additionally, we assume that the adversary can choose a worst-case target sample as is standard for auditing DP mechanisms [20]. Finally, we assume that the adversary can choose (possibly adversarial) initial model parameters. Note that this is not only an assumption allowed by DP-SGD's privacy analysis, but, in the context of the popular public pre-training, private fine-tuning regime [11], this is also a reasonable assumption.

**Auditing.** Our DP-SGD auditing procedure is outlined in Algorithm 2. As we operate in the black-box threat model, the procedure only makes a call to the DP-SGD algorithm—once again, they cannot insert canary gradients, etc. As mentioned, the adversary can set the initial model parameters ($\theta_0$), which is fixed for all models. Finally, we assume that the final model parameters are given to the adversary, who calculates the loss of the target sample as "observations." These are then thresholded to compute FPRs and FNRs and estimate the $\varepsilon_{emp}$ using the EstimateEps process (see Section 3.3). This threshold, which we denote as $\tau$, must be computed on a separate set of observations (e.g., a *validation* set) for the $\varepsilon_{emp}$ to constitute a technically valid lower bound. However, as any decision threshold for GDP is equally likely to maximize $\varepsilon_{emp}$ [28], we follow common practice [27, 29, 28] to find the threshold that maximizes the value of $\varepsilon_{emp}$ from $O \cup O'$.

**Crafting worst-case initial parameters.** Recall that DP-SGD's privacy analysis holds not only for randomly initialized models but also for *arbitrary* fixed parameters. Indeed, this was noted in prior work [11, 20, 28] and led to using initial model parameters that are fixed for all models. In our work, we consider *worst-case*, rather than *average-case*, initial parameters that minimize the gradients of normal samples in the dataset. This makes the target sample's gradient much more distinguishable. To do so, we pre-train the model using non-private SGD on an auxiliary dataset that follows the same distribution as the target dataset. Specifically, for MNIST, we pre-train the model on half of the full dataset (reserving the other half to be used for DP-SGD) for 5 epochs with a batch size of 32 and a learning rate of 0.01. For CIFAR-10, we first pre-train the model on the CIFAR-100 dataset for 300 epochs with batch size 128 and a learning rate of 0.1. Then, we (non-privately) fine-tune the model on half of the full dataset for 100 epochs, with a batch size of 256 and a learning rate of 0.1. Ultimately, we show that the gradients of normal samples are indeed minimized, achieving significantly tighter audits of DP-SGD in the black-box model.

# 5 Experiments

In this section, we evaluate our auditing procedure using image datasets commonly used to benchmark differentially private classifiers and CNNs that achieve reasonable utility on these datasets.

| Dataset | $\varepsilon = 1.0$ | $\varepsilon = 2.0$ | $\varepsilon = 4.0$ | $\varepsilon = 10.0$ |
|---------|---------|---------|---------|----------|
| MNIST | 95.4 | 95.7 | 95.9 | 95.9 |
| CIFAR-10 | 46.0 | 51.2 | 52.2 | 53.6 |

Table 1: Accuracy of models for average-case initial model parameters ($\theta_0$).

Due to computational constraints, we only train $2R = 200$ models (100 with the target sample and 100 without) to determine the empirical privacy estimates $\varepsilon_{emp}$ in the full model training setting and train $2R = 1000$ models for the last-layer only fine-tuning setting. In theory, we could achieve tighter audits by increasing the number of models, however, this would require significantly more computational resources and time (see below). Since our work focuses on the (broader) research questions of whether tight audits are possible in the black-box setting and how various factors affect this tightness, we believe $2R = 200$ provides a reasonable tradeoff between tightness and computational overhead.

In our evaluation, we report lower bounds with 95% confidence (Clopper-Pearson [10]) and report the mean and standard deviation values of $\varepsilon_{emp}$ over five independent runs.[2]

## 5.1 Experimental Setup

**Datasets.** We experiment with the MNIST [24] and CIFAR-10 [23] datasets. The former includes 60,000 training and 10,000 testing 28x28 grayscale images in one of ten classes (hand-written digits), Often referred to as a "toy dataset," is commonly used to benchmark DP machine learning models. CIFAR-10 is a more complex dataset containing 50,000 training and 10,000 testing 32x32 RGB images, also consisting of ten classes. Note that the accuracy of DP models trained on CIFAR-10 has only recently started to approach non-private model performance [11, 35]. Whether on MNIST or CIFAR-10, prior black-box audits (using average-case initial parameters) have not been tight. More precisely, for theoretical $\varepsilon = 8.0$, De et al. [11] and Nasr et al. [28] only achieve empirical estimates of $\varepsilon_{emp} \approx 2$ and $\varepsilon_{emp} \approx 1.6$, respectively, on MNIST and CIFAR-10.

To ensure a fair comparison between the average-case and worst-case initial parameter settings, we split the training data in two and privately train on only half of each dataset (30,000 images for MNIST and 25,000 for CIFAR-10). The other half of the training dataset is used as the auxiliary dataset to non-privately pre-train the models and craft the worst-case initial parameters (see Section 4). We note that for CIFAR-10, we additionally augment the pre-training dataset with the CIFAR-100 dataset to boost the model accuracy further (see Appendix A for further details).

**Models.** When training the model fully, we use a (shallow) *Convolutional Neural Network (CNN)* for both datasets. We do so as we find accuracy to be poor with models like Logistic Regression and Fully Connected Neural Networks, and training hundreds of deep neural networks is impractical due to large computational cost. Therefore, we draw on prior work by Dörmann et al. [14] and train shallow CNNs that achieve acceptable model accuracy while being relatively fast to train. (We report exact model architectures in Appendix A). Whereas, for the last layer-only fine-tuning setting, we follow [11] and use the *Wide-ResNet (WRN-28-10)* [41] model pre-trained on ImageNet-32 [12].

**Experimental Testbed.** All our experiments are run on a cluster using 4 NVIDIA A100 GPUs, 64 CPU cores, and 100GB of RAM. Auditing just one model took approximately 16 GPU hours on MNIST and 50 GPU hours on CIFAR-10.

## 5.2 Full Model Training

To train models with an acceptable accuracy on MNIST and CIFAR-10, we first tune the hyper-parameters (i.e., learning rate $\eta$ and number of iterations $T$); due to space limitations, we defer discussion to Appendix A.

**Model Accuracy.** In Table 1, we report the accuracy of the final models for the different values of $\varepsilon$-s we experiment with. While accuracy on MNIST is very good ($\geq 95\%$ for all $\varepsilon$), for CIFAR-10,

---

[2]The source code needed to reproduce our experiments is available from `https://github.com/spalabucr/bb-audit-dpsgd`.

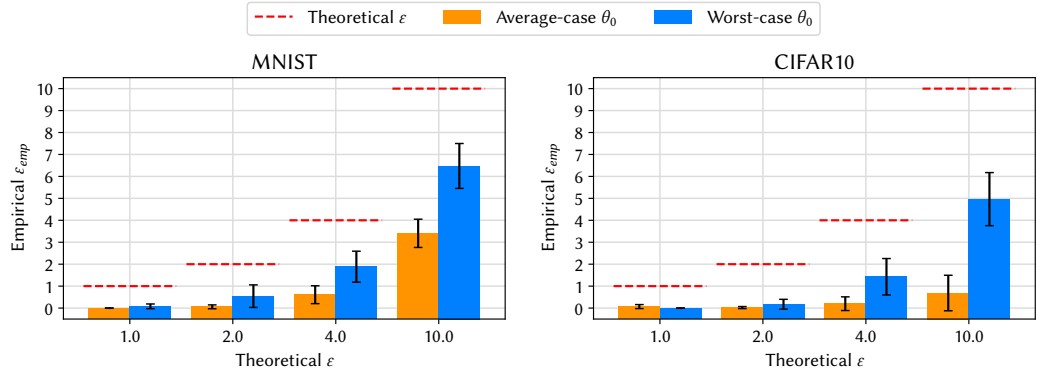

Figure 1: Auditing models with average-case vs worst-case initial parameters at various levels of $\varepsilon$.

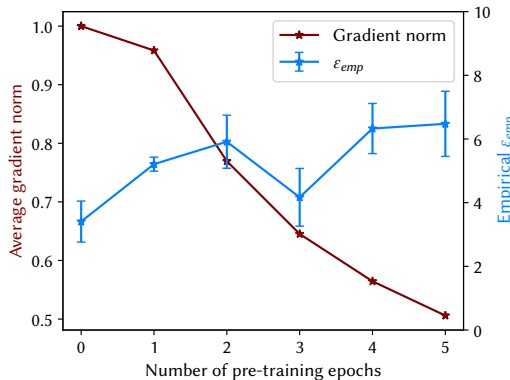

Figure 2: Comparing the average gradient norms to empirical privacy leakage, $\varepsilon_{emp}$, for models trained on MNIST at $\varepsilon = 10.0$ with increasing number of pre-training epochs.

we are relatively far from the state of the art ($\approx 50\%$ at $\varepsilon = 10.0$ vs. $\approx 80\%$ at $\varepsilon = 8.0$ in [11]). This is due to two main reasons. First, as we need to train hundreds of models, we are computationally limited to using shallow CNNs rather than deep neural networks. Second, we need to consider a much larger batch size to ease auditing [28]; thus, the models converge much more slowly. In Appendix B, we show that, given enough iterations ($T = 1,000$), the model reaches an accuracy of $\approx 70\%$ at $\varepsilon = 10.0$, almost approaching state-of-the-art. As a result, we audit models striking a reasonable trade-off between utility and computational efficiency, as discussed above.

**Impact of initialization.** In Figure 1, we compare the difference in $\varepsilon_{emp}$ between worst-case initial parameters and average-case parameters (as done in prior work [11, 20, 28]). In the latter, all models are initialized to the same, randomly chosen $\theta_0$ (using Xavier initialization [18]), while, in the former, to a $\theta_0$ from a model pre-trained on an auxiliary dataset. In the rest of this section, the target sample is set as the blank sample, which is known to produce the tightest audits in previous work [11, 28].

With worst-case initial parameters, we achieve significantly tighter audits for both MNIST and CIFAR-10. Specifically, at $\varepsilon = 1.0, 2.0, 4.0, 10.0$, the empirical privacy leakage estimates reach $\varepsilon_{emp} = 0.08, 0.54, 1.89, 6.48$ and $\varepsilon_{emp} = 0.00, 0.18, 1.43, 4.96$ for MNIST and CIFAR-10, respectively. In comparison, with average-case initial parameters as in prior work, the empirical privacy leakage estimates were significantly looser, reaching only $\varepsilon_{emp} = 0.00, 0.06, 0.61, 3.41$ and $\varepsilon_{emp} = 0.07, 0.02, 0.20, 0.69$ for MNIST and CIFAR-10, respectively. Notably, at larger privacy levels $\varepsilon = 4.0$ and $\varepsilon = 10.0$, for both MNIST and CIFAR-10, the empirical privacy leakage estimates when auditing with worst-case initial parameters are not only larger than the average-case, but are outside of the $\pm$ s.d. range. This shows that our auditing procedure not only produces significantly tighter audits than prior work, especially at larger $\varepsilon$s, but also that, at lower $\varepsilon$-s, our evaluations are currently limited by the number of models used to audit (resulting in large s.d.).

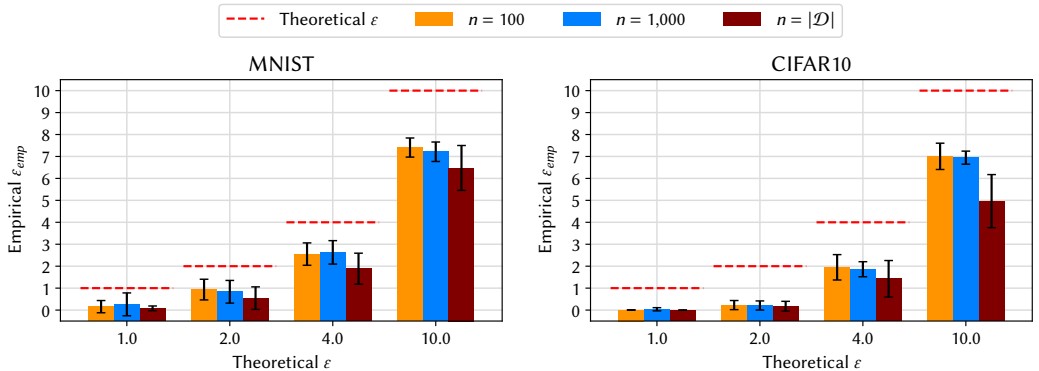

Figure 3: Auditing models trained on varying dataset sizes ($n$) at different values of $\varepsilon$. The full dataset size $|\mathcal{D}|$ is 30,000 for MNIST and 25,000 for CIFAR-10.

To shed light on why worst-case initial parameters improve tightness, we vary the amount of pre-training done to craft $\theta_0$ and, in Figure 2, plot the corresponding $\varepsilon_{emp}$-s and the average gradient norm of all samples (after clipping) in the first iteration of DP-SGD. For computational efficiency, we focus on MNIST and set the theoretical $\varepsilon$ to 10.0. Recall that, regardless of the initial model parameters, all models achieve $\geq 95\%$ accuracy on the test set. As the number of pre-training epochs increases from 0 (no pre-training) to 5, the average gradient norm steadily decreases from 1.00 to 0.51, as expected. On the other hand, the empirical privacy leakage increases from 3.41 to 6.48. In other words, with more pre-training, the impact of the *other* samples in the dataset reduces, making the *target* sample much more distinguishable, thus yielding tighter audits. Although there appears to be an anomaly in the $\varepsilon_{emp}$ obtained when the model is pre-trained for 3 epochs, this is within the standard deviation.

*Remarks.* With lower $\varepsilon$ values, the empirical $\varepsilon_{emp}$ estimates are still relatively far away from the theoretical bounds. This is because, in each run, we train a small number of models and report the *average* $\varepsilon_{emp}$ across all five runs. In fact, the *maximum* $\varepsilon_{emp}$ obtained over the five runs amount to $\varepsilon_{emp} = 0.27, 1.48, 3.23, 8.28$ and $\varepsilon_{emp} = 0.00, 0.54, 2.33, 6.70$, respectively, on MNIST and CIFAR-10, for $\varepsilon = 1.0, 2.0, 4.0, 10.0$. Nevertheless, even on the averages, our audits are still appreciably tighter than prior work—approximately by a factor of 3 for MNIST [11] and CIFAR-10 [28]. Overall, this confirms that the tightness of the audits can vary depending on the datasets, possibly due to differences in the difficulty of the classification task on the datasets. For instance, MNIST classification is known to be easy, with DP-SGD already reaching close to non-private model performance, even at low $\varepsilon$-s. By contrast, CIFAR-10 is much more challenging, with a considerable gap still existing between private and non-private model accuracy [11].

**Impact of size of dataset.** In the auditing literature, empirical privacy leakage is often commonly estimated on (smaller) dataset samples both to achieve tighter audits and for computational efficiency reasons [11]. In Figure 3, we evaluate the impact of the dataset size ($n$) on the tightness of auditing. While smaller datasets generally yield tighter audits, the size's impact also depends on the dataset itself. For instance, at $\varepsilon = 10.0$, auditing with $n = 100$ results in $\varepsilon_{emp} = 7.41$ and $7.01$, for MNIST and CIFAR-10, respectively, compared to $\varepsilon_{emp} = 6.48$ and $4.96$ on the full dataset. In other words, with MNIST, the dataset size does not significantly affect tightness. However, for CIFAR-10, smaller datasets significantly improve the audit's tightness significantly.

**Sensitivity to clipping norm.** Finally, we evaluate the impact of the hyper-parameters on the tightness of black-box audits. In the black-box setting, unlike in white-box, the adversary cannot arbitrarily insert gradients that are of the same scale as the gradient clipping norm, $C$ [28]. Thus, they are restricted to gradients that naturally arise from samples. In Figure 4, we report the $\varepsilon_{emp}$ values obtained with varying clipping norms, $C$. For computational efficiency, the results on CIFAR-10 are for models trained on datasets of size $n = 1,000$.

When the gradient clipping norm is small ($C = 0.1, 1.0$), the audits are tighter. This is because the gradient norms are typically very small in the worst-case initial parameter setting, leading to a much higher "signal-to-noise" ratio. While this might suggest that black-box audits may not be as

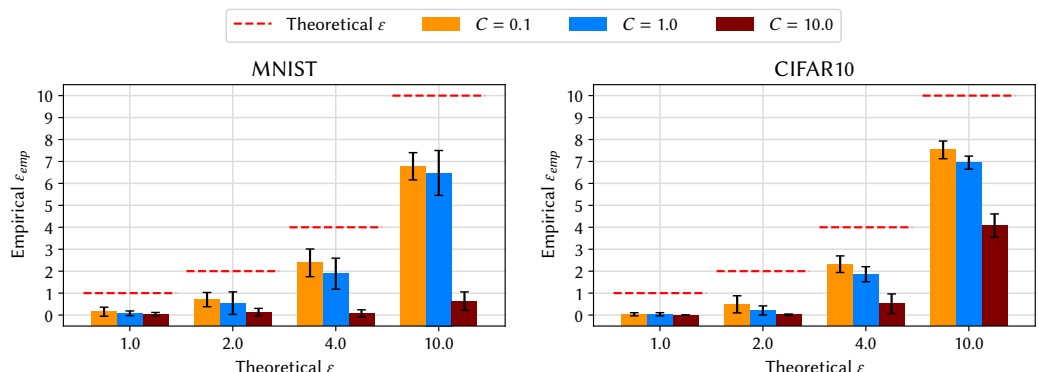

Figure 4: Auditing models trained with varying gradient clipping norm ($C$) at $\varepsilon = 1.0, 2.0, 4.0, 10.0$.

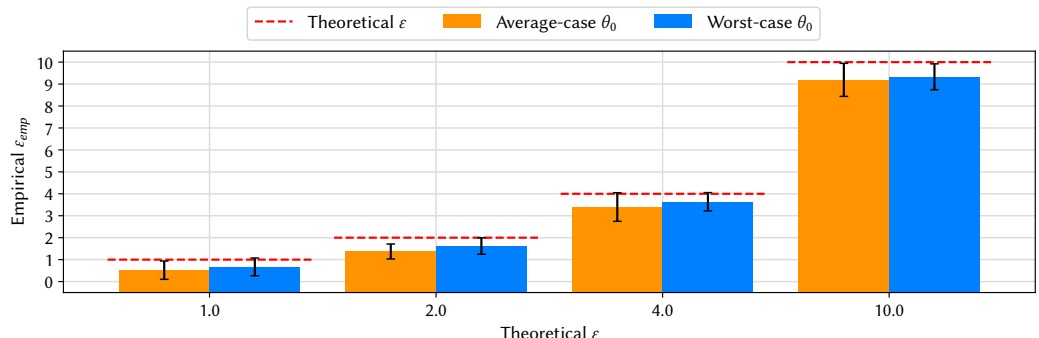

Figure 5: Auditing models when fine-tuning only the last layer using average-case and worst-case initialization of models at various levels of $\varepsilon$.

effective with larger gradient clipping norms, we observe that, with $C = 10.0$, model accuracy is poor too—more precisely, with $\varepsilon = 1.0, 2.0, 4.0, 10.0$, accuracy reaches $85.6\%, 93.1\%, 95.0\%, 96.1\%$, respectively, compared to $97.2\%, 97.4\%, 97.5\%, 97.5\%$ with $C = 1.0$ on MNIST. This indicates that audits are looser with $C = 10.0$ because the model is not learning effectively, as there is too much noise. In fact, this could suggest that the privacy analysis for the $C = 10.0$ setting could be further tightened in the black-box threat model, which we leave to future work.

### 5.3 Fine-Tuning the Last Layer Only

We now experiment with the setting where a model is first non-privately pre-trained on a large public dataset, and *only the last layer* is fine-tuned on the private dataset. For this setting, we choose the WRN-28-10 model pre-trained on ImageNet-32 and fine-tune the last layer on CIFAR-10 using DP-SGD, as done in [11]. At $\varepsilon = 1.0, 2.0, 4.0, 10.0$, fine-tuning the last layer yields model accuracies of $91.3\%, 91.4\%, 91.4\%, 91.5\%$ and $93.1\%, 93.2\%, 93.3\%, 93.2\%$ for average-case and worst-case initial model parameters, respectively. In this section, we run the ClipBKD procedure from [20] to craft a target sample, as it has been shown to produce tighter empirical privacy guarantees for Logistic Regression models.

Once again, we compare the tightness of the audits with respect to the initial model parameters. In Figure 5, we report the empirical privacy leakage estimates obtained at various levels of $\varepsilon$ when the last layer is initialized to average- and worst-case (through pre-training) initial model parameters. Similar to the full model training setting, model initialization can impact audit tightness, albeit not as significantly. Specifically, at $\varepsilon = 10.0$, we obtain $\varepsilon_{emp} = 9.19$ and $9.33$, respectively, when the last layers are initialized to average- and worst-case initial model parameters.

Overall, the audits in this setting are considerably tight, and increasingly so for larger $\varepsilon$, as we obtain empirical privacy leakage estimates of $\varepsilon_{emp} = 0.67, 1.62, 3.64, 9.33$ for $\varepsilon = 1.0, 2.0, 4.0, 10.0$,

respectively. The maximum $\varepsilon_{emp}$ across five independent runs is also much tighter, namely, $\varepsilon_{emp} = 1.37, 2.10, 4.28, 9.94$ for $\varepsilon = 1.0, 2.0, 4.0, 10.0$, respectively. Although maximum $\varepsilon_{emp}$ exceeds the theoretical $\varepsilon$ at 1.0, 2.0, and 4.0 suggests a DP violation, this is, in fact, within the standard deviation of $\pm 0.40, 0.36$, and $0.38$, respectively.

We also evaluated the impact of the dataset size and clipping norm in this setting as well; however, these factors do not significantly impact audit tightness, as this is a much simpler setting compared to training a CNN model fully. For more details, please see Appendix C.

## 6    Conclusion

This paper presented a novel auditing procedure for Differentially Private Stochastic Gradient Descent (DP-SGD). By crafting worst-case initial model parameters, we achieved empirical privacy leakage estimates substantially tighter than prior work for DP-SGD in the black-box model and for natural (i.e., not adversarially crafted) datasets. At $\varepsilon = 10.0$, we achieve nearly tight estimates of $\varepsilon_{emp} = 7.21$ and 6.95 for datasets consisting of 1,000 samples from MNIST and CIFAR-10, respectively. While we achieve slightly weaker estimates of $\varepsilon_{emp} = 6.48$ and $4.96$ on the full datasets, these still represent a roughly 3x improvement over the estimates achieved in prior work [11, 28].

Naturally, our work is not without limitations – the main one being the computational cost of auditing. Black-box auditing typically requires training hundreds of models to empirically estimate FPRs/FNRs with good accuracy and confidence. This can take hundreds of GPU hours, even for the shallow CNNs we audit in our work. Thus, auditing deep neural networks (e.g., WideResNet) trained on large datasets (e.g., ImageNet) may be computationally challenging for entities that are not huge corporations. However, as our main objective is addressing the open research question of whether tight DP-SGD audits are feasible in the black-box model, we focus on shallow models. Our results are very promising as, thanks to the intuition of using worst-case initial parameters, we do achieve nearly tight audits. Nevertheless, we leave it to future work to reduce the computational cost of the auditing procedure, e.g., by training significantly fewer models.

Relatedly, recent work [2, 34] has begun to focus on auditing within a single training run, although it has thus far achieved only loose empirical estimates. One interesting direction would be to combine the insights from this paper (i.e., using worst-case initial model parameters) with one-shot auditing techniques. However, this may not be trivial as these techniques typically employ a large number of canary samples, each with large gradient norms, which can potentially interfere with our goal to reduce the gradient norm of "other" samples.

Furthermore, we only consider full batch gradient descent ($B = n$), i.e., without sub-sampling. Note that this is standard practice and eases auditing by enabling GDP auditing, which requires fewer models and is much less computationally intensive compared to auditing DP-SGD with sub-sampling using Privacy Loss Distributions (PLDs) [28]. Nevertheless, when auditing DP-SGD with sub-sampling in the "hidden-state" threat model where the adversary can only observe the final trained model but can insert *gradient canaries*, recent work has shown that there is a significant gap between the theoretical upper bound and the empirical lower bound privacy leakage achieved [8]. Even though this suggests a privacy amplification phenomenon in the hidden-state setting (which would extend to black-box as well), for general non-convex loss functions, prior work has also shown that the privacy analysis of DP-SGD with sub-sampling is tight even in the hidden-state [3]. Therefore, whether audits can be tight in the black-box threat model for DP-SGD with sub-sampling under realistic loss functions and datasets remains an open question for future work.

### Acknowledgments and Disclosure of Funding

This work has partly been supported by a National Science Scholarship (PhD) from the Agency for Science Technology and Research, Singapore (A*STAR). We also wish to thank Jamie Hayes for providing ideas and feedback throughout the project.

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

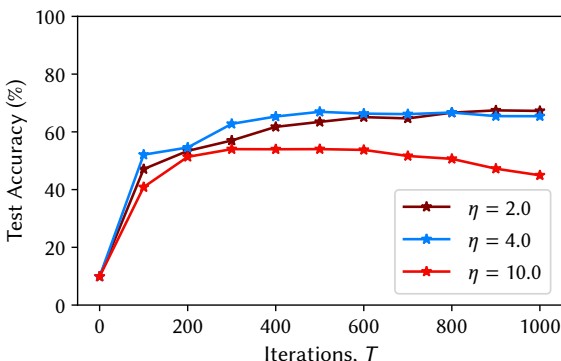

Figure 6: Test accuracies (%) on CIFAR-10 for models trained for varying number of iterations, $T$ and learning rates, $\eta$ at a fixed $\varepsilon = 10.0$.

## A  Model Architectures and Hyper-parameters

In this work, we use the shallow Convolutional Neural Networks used by Dörmann et al. [14], who make minor modifications to the CNNs used earlier by Tramèr and Boneh [35] to achieve good model utilities on MNIST and CIFAR-10. Exact model architectures for MNIST and CIFAR-10 are reported in Tables 2 and 3, respectively.

| Layer | Parameters |
| --- | --- |
| Convolution | 16 filters of 5x5 |
| Max-Pooling | 2x2 |
| Convolution | 32 filters of 4x4 |
| Max-Pooling | 2x2 |
| Fully connected | 32 units |
| Fully connected | 10 units |

Table 2: Shallow CNN model for MNIST with Tanh activations.

| Layer | Parameters |
| --- | --- |
| Convolution | 32 filters of 3x3, stride 1, padding 1 |
| Convolution | 32 filters of 3x3, stride 1, padding 1 |
| Max-Pooling | 2x2, stride 2, padding 0 |
| Convolution | 64 filters of 3x3, stride 1, padding 1 |
| Convolution | 64 filters of 3x3, stride 1, padding 1 |
| Max-Pooling | 2x2, stride 2, padding 0 |
| Convolution | 128 filters of 3x3, stride 1, padding 1 |
| Convolution | 128 filters of 3x3, stride 1, padding 1 |
| Max-Pooling | 2x2, stride 2, padding 0 |
| Fully connected | 128 units |
| Fully connected | 10 units |

Table 3: Shallow CNN model for CIFAR-10 with Tanh activations

**Hyper-parameter tuning.** To achieve the best model utilities, we tune the hyper-parameters of DP-SGD (i.e., number of iterations, $T$ and learning rate, $\eta$). For MNIST, we train for $T = 100$ iterations, with a learning rate of $\eta = 4$ for both average-case and worst-case initial model parameters. For CIFAR-10, we train for $T = 200$ iterations, with a learning rate of $\eta = 2$ for the average-case initial model parameter setting. For the worst-case initial model parameter setting, we use a learning rate of $\eta = 1$ instead, as this resulted in the best model performance for CIFAR-10. When fine-tuning just the last layer, we train for $T = 25$ iterations, with a learning rate of 10.0. Note that, for the experiments varying the dataset size, the learning rate was scaled accordingly so that the size of the dataset does not affect the "influence" of each sample (i.e., $\frac{\eta}{B}$). All clipping norms are set to $C = 1.0$ as done in prior work [11], and batch sizes are set to the dataset size $B = n$ to ease auditing [28]. Finally, the noise multiplier $\sigma$ is calculated from the batch size and number of iterations using the Privacy loss Random Variable accountant provided by Opacus [40].

## B  Model Convergence

In Figure 6, we plot the test accuracies obtained when training the CNN model on the CIFAR-10 dataset for varying number of iterations, $T$, and learning rates, $\eta$, at a fixed theoretical $\varepsilon = 10.0$. Specifically, we train with full batch gradient descent (i.e., $B = n$) and re-calculate the noise

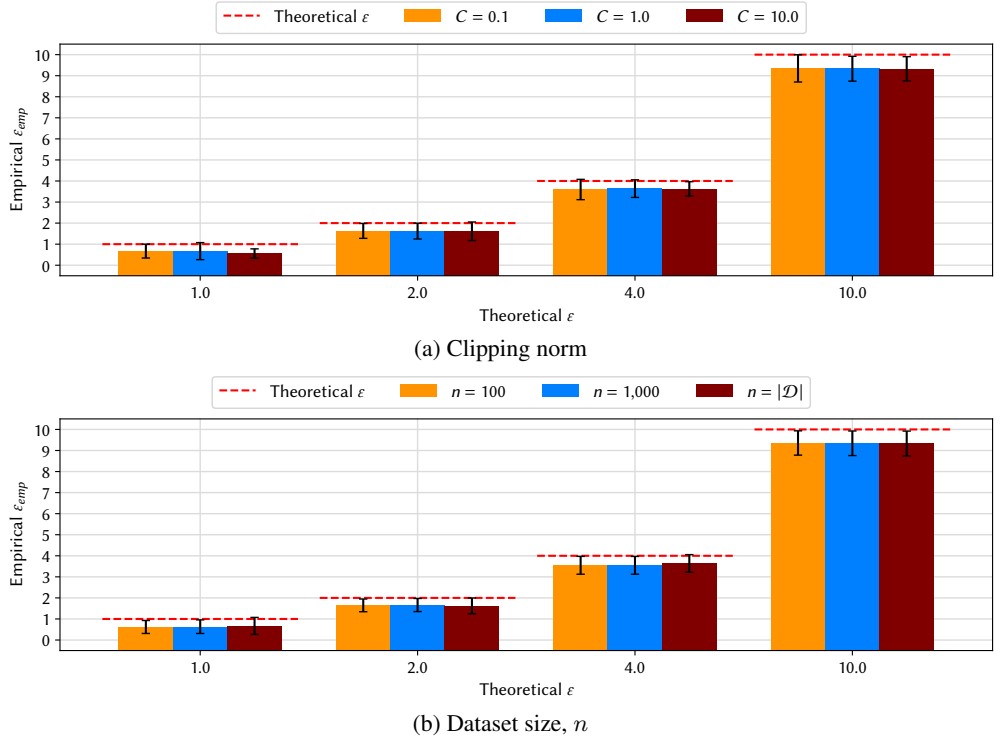

(a) Clipping norm

(b) Dataset size, $n$

Figure 7: Comparing the tightness of $\varepsilon_{emp}$ for different factors at different values of $\varepsilon$

multiplier $\sigma$ for each $T$ and $\eta$ such that all points on the Figure have the same privacy level of $\varepsilon = 10.0$.

First, we observe that $\eta = 2.0$ and $\eta = 4.0$ result in comparable test accuracies that increase with the number of iterations, reaching close to 70% accuracy at $T = 1,000$ (67.3% and 65.4%, respectively). Therefore, even though at $T = 200$ and $\varepsilon = 10.0$, the models we audit only achieve 53.6% accuracy, this is a *computational* limitation, and the model can, in fact, achieve good model utilities given enough iterations. However, increasing the learning rate to $\eta = 10.0$ results in poor model performance.

In their original work, Dörmann et al. achieve a test accuracy of 70.1% at $\varepsilon = 7.42$ by training the model with a batch size of 8,500 (sampling rate, $q = 0.17$). However, following prior work [28], we choose not to consider sub-sampling as this makes tight auditing difficult. This is because the theoretical privacy region of the sub-sampled Gaussian mechanism (the underlying mechanism for DP-SGD with sub-sampling) is loose compared to the empirical privacy region observed, thus making audits in this setting inherently weaker.

## C  Last layer only fine-tuning

Finally, we evaluate the impact of the clipping norm and dataset size on the tightness of auditing in the last layer only fine-tuning setting. Overall, we find that these factors have little to no impact, as tight audits are achieved in the most difficult settings considered.

In Figures 7a and 7b, we report the empirical privacy estimates obtained when using different gradient clipping norms and when training on datasets with different sizes, respectively. While there may be some minor differences between the empirical $\varepsilon_{emp}$ obtained in different settings, overall, the gradient clipping norm and dataset size do not significantly affect tightness. We believe this is because it is much simpler to fine-tune the last (Logistic Regression) layer of a CNN than to train the full model. Here, the empirical privacy leakage is already maximized ($\varepsilon_{emp} \approx 10.0$ for theoretical $\varepsilon = 10.0$) in the most difficult setting ($C = 10.0$, $n = |\mathcal{D}|$). Therefore, "relaxing the setting" by using smaller clipping norms and dataset sizes does not significantly improve the privacy leakage.

