# OpenReview forum: "Nearly Tight Black-Box Auditing of Differentially Private Machine Learning"
_NeurIPS.cc/2024/Conference — NeurIPS 2024 poster_

### Official Review · Reviewer_JVDo · 2024-07-09

**Soundness:** 2
**Presentation:** 3
**Contribution:** 2
**Rating:** 5
**Confidence:** 3

**Summary:**

This paper studies the problem of auditing DP-SGD in the black box threat model, i.e. only black-box access to the last iterate $\theta_T$ of DP-SGD. The paper's main contribution is to show experimentally that pre-training helps to get tighter auditing. The argument behind using this idea is that pre-training makes the average gradient norm of “in-distribution” points small, thus making the target point used for auditing/membership attack more “distinguishable”. The paper explores this worst-case initialisation idea for two datasets, MNIST and CIFAR10, and reports tighter privacy lower bounds.

**Strengths:**

- Well-motivated problem. Tight black-box auditing is still an open problem compared to the white-box setting.
- Extensive experimental exploration of the effect of different hyperparameters on the auditing procedure.
- The paper provides confidence intervals on the empirical privacy budget $\epsilon$.

**Weaknesses:**

- The main contribution of this work is incremental. In addition, as noted in the paper, the idea that the initialisation of $\theta_0$ affects auditing/membership inference has already been reported in the literature [10,18,26].

- The justification for using pre-training is hand-wavy and not rigorous. See the question below.

- No code is available to reproduce the results. The experimental setting is well-detailed. However, some important factors for reproducing the results have not been reported. See questions below.

**Questions:**

- Worst-case vs average-case initialisation: (a) For the worst-case, $\theta_0$ is selected by pretraining on half of the dataset, but what algorithm is used for that? Is it non-private SGD? And what is the random distribution used to generate $\theta_0$ for the average case?

- In Figure 5, the average-case initialisation seems to give tighter privacy and lower bounds, especially for small epsilon. This contradicts this work's main argument, i.e., using worst-case initialization is better. How do you explain this?

- What canary strategy is used for the black-box audit? Do you only use the blank example? Did you test any other black box canaries?

- The justification behind the reported success of pretraining is that the average gradient norm decreases with the number of pre-training epochs, reported in Figure 2. (a) Is the average of gradients computed on all samples (training+ test) or only training samples? (b) Why does this correlate with the success of your membership attack? Specifically, the MIA attack tresholds over the loss of the canary, thus this seems to be the statistic that should be analysed over the pre-training epochs. The gradient average norm is indicative for white-box attacks. (c) Pretraining decreases the gradient average loss of “in distribution” samples, so how would you adapt your auditing procedure for datasets where the empty sample, i.e. your canary, is "in distribution"?

- What are the loss thresholds used for the black-box membership attack? Are they fine-tuned to find the best privacy budget? Also, the function EstimateEps() is never explicitly detailed in the paper to reproduce the results.

- Is it not already possible to use [32]’s framework to run directly your black-box auditing in one run?

**Limitations:**

As noted by the authors in the conclusion, the auditing procedure is computationally heavy and time expensive.

---

> ### Author Rebuttal · Authors · 2024-08-02
>
> We thank reviewer JVDo for their helpful and constructive feedback. In the following, we address their comments and questions and clarify how we believe we can address their concerns regarding weaknesses.
>
> > **Contributions of crafting worst-case initial model parameters**
>
> Although the general idea of looking at initialization was indeed already reported in [10, 18, 26], these works only looked at tweaking the randomization of initialization. Instead, we focus on specifically crafting worst-case initial model parameters, which had not been considered before. We refer the Reviewer to Point 1. of the Global Author Rebuttal comment for a more detailed comparison with prior work.
>
> > **Experiments’ details**
>
> We thank the reviewer for highlighting the gaps in the details of our experimental analysis. In addition to addressing the reviewer’s questions below, please note that we will make the code publicly available with the camera ready; in the meantime, we would be happy to share the code anonymously upon request.
>
> **Answers to Questions**
>
> > **Worst-case vs Average-case initialization**
>
> We refer the Reviewer to Point 3. of the Global Author Rebuttal comment for details regarding crafting worst-case initial model parameters (i.e., pre-training on half of the dataset). For the random distribution, we follow [18] and use Xavier initialization, which we will clearly explain in Section 5.
>
> > **Discrepancies in Figure 5**
>
> We note that the discrepancy is due to the small number of models and refer the Reviewer to the Point 2. of the Global Author Rebuttal comment, where we explain this in detail.
>
> > **Canary strategy**
>
> We use the blank sample when auditing the full model and the ClipBKD sample [18] when auditing the last-layer only fine-tuning. We experimented with both samples (along with a “one-hot” sample) and used the sample with the tightest audits for both settings. We note that ClipBKD performed better for last-layer only fine-tuning as it was particularly designed and tested for this setting. We will clarify this in the revised version of the paper in Section 5.1.
>
> > **Success of pre-training strategy**
>
> In Figure 2, (a) the average of the gradient norms is only computed on training samples. (b) The gradient norms of other samples are important to the success of our attack as they represent the expected change in the loss function due to the other samples during training. Specifically, when the gradient norms of other samples are close to 0, the loss function is mainly impacted by the presence of the target sample only, making the loss of the target much more distinguishable (when the target sample is present vs. when it is not). Put simply, the gradient norms indicate the level of “noise” in the dataset compared to the “signal.” Note that while prior work [18, 26] does aim to maximize the loss of the target sample by carefully crafting the target sample, this has not been enough to achieve tight audits. In our work, we find that we are able to explain (part of) this gap by using the gradient norms of the other sample taken as “noise.” We hope this provides additional intuition as to why our approach works and will clarify this in Section 5.2 as well. (c) Even if the empty sample is “in distribution,” we can mislabel the sample to make it “out of distribution” and use the mislabelled sample as the target sample, which is also a common method used in prior work.
>
> > **Loss thresholds**
>
> As mentioned in Section 4, the loss thresholds are indeed fine-tuned to maximize privacy leakage. We will explicitly include the EstimateEps function (in the main body if there is space and in the appendix otherwise) and also release the code publicly for reproducibility.
>
> > **Using [32] for auditing in a single run**
>
> We thank the reviewer for making the point that our method can potentially be combined with [32] to audit using a single training run, as also briefly mentioned in Section 6. However, this may not be trivial as a large number of target samples with potentially large gradient norms are used in [32]. This might interfere with our approach to reduce the contribution of other samples. Therefore, we leave this to a future paper to investigate this approach in more detail. In the revised version of the paper, we will additionally state these challenges of combining our work with [32].

---

> > ### Comment · Reviewer_JVDo · 2024-08-07
> >
> > I would like to thank the authors for addressing most of my concerns. I updated my score accordingly.

---

> > > ### Author Response · Authors · 2024-08-13
> > >
> > > We once again thank reviewer JVDo for their time and updating their score to reflect their concerns being addressed.

---

### Official Review · Reviewer_uyKT · 2024-07-10

**Soundness:** 4
**Presentation:** 3
**Contribution:** 3
**Rating:** 6
**Confidence:** 3

**Summary:**

The paper considers the problem of auditing DP-SGD, i.e. deriving empirical lower bounds on the DP parameter $\epsilon$. Many previous auditing works operated in the white-box setting, i.e. are allowed to choose arbitrary gradients for the canary (sensitive example) during DP-SGD, or were in the black-box setting (i.e. could determine some aspects of training, but not specific per-round gradients, and also does not get to see intermediate states) but had a sizeable gap between the empirical $\epsilon$ and the theoretical upper bound. The paper gives a black-box auditing procedure that in some cases achieves empirical $\epsilon$ very close to the theoretical upper bound on standard benchmarks of MNIST and CIFAR10. The main algorithmic insight is to use a model pre-trained on in-distribution data, which causes the gradients of in-distribution examples to be quite small even in the first round of private training, hence increasing the impact of the canary on training. In contrast, previous work used 'average-case' initializations without optimizing for their impact on the empirical $\epsilon$. The authors perform a number of experiments using this improvement, on MNIST and CIFAR, varying whether they train a full model vs the last layer, and varying the dataset size and clip norm. The authors' audit generally improves upon the baseline of an average-case initialization, sometimes as much as a 5x increase in the empirical $\epsilon$.

**Strengths:**

* The paper provides the first audits which come close to matching the theoretical upper bounds while operating in a black-box setting. In particular the empirical improvement over past black-box work is incredibly strong, and the qualitative improvement over white-box audits is important given the threat model the authors use is much more realistic.
* The empirical results are especially impressive given the number of training runs the authors use (which is much smaller than some other auditing works) and the fact that the change needed to achieve the increase in empirical $\epsilon$ is relatively lightweight (a single pre-training run).
* The authors do a good job explaining intuition for why changing different aspects of the training pipeline affects the empirical $\epsilon$ which makes it more likely others can build upon this intuition in practice or in future work, and the intuition is reflected in the empirical results.

**Weaknesses:**

* In some cases we may want to do privacy auditing not to e.g. find bugs in an existing DP-SGD pipeline, but to support the claim that a model has much better privacy guarantees than the worst-case theoretical $\epsilon$ we are reporting. In this case, taking half of the dataset and using it as public pre-training data a privacy violation, i.e. it is unclear how to extend the authors' insights to this application of auditing. This might be a limited weakness, as anyway in practice we are probably going to try to pre-train on a distribution as close to the private distribution as possible, which might retrieve the benefits the authors observe.
* The improvements in Figure 5 don't seem to be very significant, and for eps = 1.0 the average empirical epsilon of average-case initialization is in fact double that of worst-case initialization. This did not affect my score as I understand the authors may not have access to enough compute budget to narrow the confidence intervals, and also I don't think they should be punished for reporting insignificant results, but I think the improvement here is somewhat overstated in the paper.

**Questions:**

* Do the authors believe the results in Figure 5 would become significant with more trials? i.e. do the authors have some intuition for why an initialization might matter more or less for last-layer-only fine-tuning than for full-model training? If so, it might be nice to include this in the paper.

**Limitations:**

Yes

---

> ### Author Rebuttal · Authors · 2024-08-02
>
> We thank reviewer uyKT for their helpful feedback. While their questions are hopefully addressed in Point 2. of the Global Author Rebuttal comment, we address their comments regarding weaknesses here.
>
> > **Pre-training on another distribution**
>
> We apologize as we are not entirely sure what the reviewer is referring to in the first point of the weaknesses in their review. We would be happy to address it if they could kindly clarify. Do they mean to say that it is unclear how our work would apply to the setting where a different dataset (instead of half of the target dataset) is used to craft the initial model parameters?
>
> If so, we believe that it would depend on how close the distributions of the other dataset are to the target dataset. Naturally, datasets with closer distributions would yield tighter audits. However, as we focused on achieving the tightest possible audits within the black-box setting in this work, we have not looked into this aspect yet. Nevertheless, we think it would be an interesting aspect to look into for future work and we will explicitly add this in Section 6.
>
> > **Insignificant improvements in Figure 5**
>
>  We believe that one of the reasons why there are not any significant improvements to the audits in this setting is that the audits are already nearly tight when using average-case initial model parameters. For instance, at theoretical $\varepsilon = 10.0$, the $\varepsilon_{emp} = 7.69$ for average-case initial model parameters already. Therefore, there is not much room for improvement in this setting resulting in comparable audits between the average-case and worst-case initial model parameters.

---

> > ### Comment · Reviewer_uyKT · 2024-08-11
> >
> > Thanks for the response.
> >
> > The first weakness is a limited one, so I do not think the authors need to stress about it. Yes, the high-level point is that having in-distribution pretraining data is not feasible in every problem setting, e.g. in the use-case of https://arxiv.org/pdf/2302.03098 who explore auditing the training pipeline in the same run used to generate the final model. I agree with the authors that for the problem setting considered in the paper, having in-distribution public data is a fine assumption.

---

> > > ### Author Response · Authors · 2024-08-13
> > >
> > > We once again thank reviewer uyKT for their time and constructive feedback. We agree with the reviewer that the first weakness is a limited one and thank the reviewer for acknowledging our rebuttal.

---

### Official Review · Reviewer_ncD1 · 2024-07-12

**Soundness:** 4
**Presentation:** 3
**Contribution:** 2
**Rating:** 6
**Confidence:** 3

**Summary:**

This paper proposes a new method for auditing DP-SGD in a black-box setting, where the auditor can only see the final parameters of the model (rather than intermediate steps). The main idea is to select worst-case initial model parameters; this seems to give a substantial advantage to the black-box analysis.

**Strengths:**

The idea of selecting worst-case initial parameters for auditing is clever. Empirically, the proposed method strongly outperforms the chosen baseline.
This work shows that a (partially) black-box adversary isn't much weaker than a white-box one; this demonstrates that DP analyses (which assume the latter) may not be underestimating privacy too much.
The paper is well-presented.

**Weaknesses:**

# Motivations
I think the authors should refine their motivations. They use two: 1) their method is good for detecting bugs, and 2) it provides insights into DP-SGD analyses.

1. Bugs.

The ultimate purpose of _black-box_ auditing is unclear. There's two sides to this discussion. On the one hand, if our goal is to audit a DP-SGD implementation, surely we should be making all the worst-case assumptions (i.e., give as much advantage to the auditor as possible) to look for bugs. On the other hand, if our goal is to evaluate real-world attacks against DP-SGD, then it does make sense to consider a black-box assumption. The assumptions this paper makes are a bit of an hybrid: it assumes black-box for auditing, yet the adversary is allowed to specify the initial model parameters. The only semi-practical scenario I can think of is an attacker who creates a model (for subsequent fine-tuning to be carried out by the victim) and wants to augment the leakage in the fine-tuned version of the model. Aside from this scenario, I would recommend leaving out from this paper any motivation arguing that this work is more practical than other auditing methods for finding "bugs"; please refer to "Questions" for a concrete example.

2. New insights.

The authors do provide one very convincing motivation: a big question is whether DP analyses of DP-SGD could be improved if they considered it as a black-box algorithm; currently, they all assume knowledge of intermediate gradients. This work offers some hope: from the analysis, it seems that the difference between black-box and the theoretically estimated "epsilon" may be closer than expected. If I may, I would strongly recommend the authors to center their motivations on this aspect.

# Comparison

I could not find a real comparison to previous auditing methods. Only two numbers appear (within the Datasets paragraph), reporting the performance of previous works; were these computed under the same experimental setting (and code base)? Ideally, they should be replicated under identical conditions to yours.
Secondly, it'd be very interesting to see what similar white-box methods achieve, in comparison. How tight are they, on those datasets/models?

**Questions:**

- "Although techniques to tightly audit DP-SGD exist in literature [26, 27], they do so only in active  white-box threat models, where the adversary can observe and insert arbitrary gradients into the  intermediate DP-SGD steps. However, real-world adversaries cannot always access the model’s  inner parameters, let alone insert arbitrary gradients."

As argued above, I don't think this is a valid premise. There's nothing in auditing that requires modelling a real-world adversary. Further, the assumptions that you make (e.g., worst-case initial parameters) do not correspond to a real-world adversary. I think the implications of what you claim here are incorrect, and I don't think it should be part of your motivations.

- Threat model. It does look weird to see a threat model section in an auditing paper, and I would humbly recommend removing it: DP doesn't specify any particular threat model, and auditing is just about ensuring how tightly the definition is being matched empirically. If you do want to keep this section, please note that it's missing the assumption that the auditor (adversary?) can specify the initial model parameters; also, I was unclear as to what the following meant in this context: "assume that the adversary can choose a worst-case target sample as is standard for auditing DP mechanisms".

- "we set B=N".

Can you please clarify whether this is a constraint of your method, and if so what are the potential limitations?

- Sentence "When auditing a DP mechanism [...]". Please define the symbol $R$. (It's clear what it is, but it's undefined.)

- "Crafting worst-case initial parameters". This should be one of the main contributions, but unfortunately the paragraph doesn't provide much information on how exactly these parameters are created. Can you please provide more information in the text (and in rebuttal)?

- Could you please double check: did your cluster only have 1 CPU core?

- Typo "is is".

**Limitations:**

These were adequately addressed.

---

> ### Author Rebuttal · Authors · 2024-08-02
>
> We thank reviewer ncD1 for their helpful feedback. In the following, we address their comments and questions.
>
> > **Motivation**
>
> Thank you very much for your comments re. our motivation. Indeed, our work is primarily focused on the “New Insights” type of motivation, i.e., determining if the analysis of DP-SGD can be improved if considered as a black-box algorithm. To that end, we will follow the Reviewer’s recommendation to center our motivation around this aspect. We will also follow the Reviewer’s recommendation to leave out any motivation that argues about the “practical” aspect of black-box auditing.
>
> > **Comparison to prior work**
>
> We note that previous work [10, 18, 26] considered the setting with subsampling (i.e., $B \neq N$) whereas we consider full batch gradient descent ($B = N$). Nevertheless, we do present a direct comparison with [10, 26], which is the “average-case $\theta_0$” setting – they, too, use the blank same and loss to carry out the audit. Although we did initially experiment with [18] (i.e., using the ClipBKD sample), we did not find that, in our setting, this significantly improved upon the results presented in “average-case $\theta_0$” (probably because [18] was only designed for and tested on Logistic Regression and Fully Connected Neural Networks) and thus left it out of the paper. In Section 5, we will make it clearer that the “average-case $\theta_0$” setting provides a direct comparison to [10, 26].
>
> **Answers to Questions**
>
> > **Threat model paragraph**
>
> We included this paragraph to delineate the differences between what was done in prior work (active white-box) and what we do (black-box). This also mirrors writing in prior papers [26, 28]. Nevertheless, we will include the worst-case initial parameters assumption in the threat model (and thank the reviewer for suggesting this).
>
> > **Worst-case target sample**
>
> We consider the setting where the rest of the dataset is chosen randomly (“natural dataset”), but the target sample is specifically chosen by the adversary and is not a “natural” sample. To make this clearer, we state that we “assume that the adversary can choose a worst-case target sample as is standard for auditing DP mechanisms.” We hope this clarifies the reviewer’s question.
>
> > **Clarifying $B = N$**
>
> We note that setting $B = N$ is not a constraint of our method; rather, it aims to make auditing easier and is done commonly [26]. Nonetheless, we note that recent work (Cebere et al., 2024) suggests that the privacy analysis when $B \neq N$ might be complicated, and even in stronger threat models, tight auditing may be difficult to achieve.
>
> > **Clarifying $R$**
>
> We will clarify in Section 3.3 that $R$ refers to the number of models/outputs.
>
> > **Details on worst-case initial model parameters**
>
> We refer the Reviewer to Point 3. of the Global Author Rebuttal comment and will clarify this further in the text.
>
> > **Typos**
>
> We apologize for the typos and will fix both of them in the text. We used 36 cores in the cluster (we will also clarify this).
>
> **References**
>
> Cebere, T., Bellet, A., & Papernot, N. (2024). Tighter Privacy Auditing of DP-SGD in the Hidden State Threat Model. arXiv:2405.14457.

---

> > ### Author Response · Authors · 2024-08-13
> >
> > We once again thank reviewer ncD1 for their time and constructive feedback. As the discussion period draws to a close, we hope our rebuttal has sufficiently addressed the reviewer's concerns and would be grateful to hear the reviewer's thoughts on our comments and clarifications.

---

### Official Review · Reviewer_qSii · 2024-07-13

**Soundness:** 3
**Presentation:** 3
**Contribution:** 3
**Rating:** 6
**Confidence:** 4

**Summary:**

The paper shows that privacy auditing is tight in the threat model where the model initialization is adversarially selected. They find that the decrease in gradient norms over the course of training helps improve the "signal to noise" ratio of the auditing example.

**Strengths:**

The paper's main finding is useful and helps the privacy auditing community get a bit closer to understanding privacy leakage.

The authors propose a reasonable hypothesis for why their approach improves on prior work.

**Weaknesses:**

The general finding that "when other examples' contributions are smaller, auditing is tighter", has been a feature of other papers on auditing, even in the adversarial dataset experiments from Nasr et al. 2021 or the design of ClipBKD in Jagielski et al 2020.

The paper's claims of "near tightness" seem somewhat exaggerated - there are several parameter settings where the proposed auditing does not even appear to be better than prior work, and all experiments show a rather large gap.

Given that the batch size is an important detail for auditing, the setting of B = n should be mentioned in the main body of the paper.

The experiments are unfortunately on rather small models, making it difficult to know how generalizable the strategy is to more recent advances in DP-ML training.

I would appreciate experiments on other datasets - several conclusions are drawn that seem dataset specific (e.g. dataset size), and it would be useful to know if there are more general trends in these.

**Questions:**

What is the gradient norm of the auditing example in your experiments? I wonder if at C=10, the gradient norm is not large enough to "max out" the full clipping norm.

Did you retune hyperparameters when changing the values of clipping norms?

Given the hypothesis that lower gradient norm of other examples leads to tighter audits, have you tried explicitly constructing an initialization where the other gradients are exactly 0? Perhaps even just nonprivate SGD on the private training data will get there.

**Limitations:**

I think the batch size should be better highlighted, but limitations related to model architectures, datasets, auditing trials are all well addressed.

---

> ### Author Rebuttal · Authors · 2024-08-02
>
> We thank reviewer qSii for their helpful feedback. In the following, we address their comments and questions.
>
> > **When other samples’ contributions are smaller, auditing is tighter**
>
> While the overarching intuitions are similar, our strategy is appreciably different from prior work [18, 27], as we also discuss in Point 1. of the Global Author Rebuttal comment vis-à-vis the novelty of crafting worst-case datasets. In the revised version of the paper, we will discuss this further to clarify how, despite the similarity of the intuitions, our work overcomes some significant challenges compared to prior work.
>
> > **Near tightness claims**
>
> Thank you for your comment. We agree that, in some settings, especially at low(er) $\varepsilon = 1.0$ and $2.0$, it is not entirely clear whether our auditing method improves upon prior work. However, we note that this is mainly due to computational limitations. In fact, there are significant improvements for larger $\varepsilon = 4.0$ and $10.0$ on both the MNIST and CIFAR-10 datasets (e.g., for MNIST, from $\varepsilon_{emp} = 3.41$ in prior work to $6.48$ in our work). This suggests that computational power is a limiting factor as lower epsilon values correspond to FPR/FNR rates that are very close to the random baseline, which would, in turn, require more models to estimate more accurately. Nevertheless, in the final version of the paper, we will tone down our claims of “near tight” audits to specify in which settings the audits are appreciably tighter than prior work.
>
> > **Stating $B = N$ in the main body**
>
> Currently, the setting of $B = N$ is stated in Section 3.2, but we will emphasize/clarify this again in Section 5.
>
> > **Generalizability to deeper models**
>
> Unfortunately, due to both the number of models required for the audit and the large batch sizes, auditing modern deep models (e.g., WideResNet) in the black-box setting tightly is computationally (very) expensive. However, our method is inherently general and does not depend on the model architecture. Furthermore, the crafting of our worst-case initial parameters is efficient as the model has to be only pre-trained once. Therefore, in theory, we do not see any reasons that would prevent the generalizability of our results other than very significant computational costs. In future work, we will look into auditing deep neural networks as soon as we have access to more compute or we can improve the efficiency of the underlying auditing techniques.
>
> > **Experiments on other datasets**
>
> Thank you for this comment. We did indeed think about additional datasets but ultimately settled on MNIST and CIFAR-10, which are considered benchmark datasets for auditing DP-SGD [10, 18, 27]. Moreover, the trends observed (e.g., smaller datasets yielding tighter audits) are the same across both MNIST and CIFAR-10 – due to the different complexities of the datasets, only the scale of the “impact” differs. Nevertheless, on Reviewer qSii’s request, we could add experiments on the FMNIST dataset as well (we are confident this will confirm the same conclusions we draw from MNIST/CIFAR-10).
>
> **Answers to Questions**
>
> > **Gradient norm of auditing sample**
>
> On average, across all iterations, the gradient norm of the auditing sample, before clipping, is $10.5$ for MNIST and $150.4$ for CIFAR-10. Therefore, we do not believe that this is a “maxing out” issue.
>
> > **Re-tuning hyper-parameters for clipping norms**
>
> The hyper-parameters were re-tuned for the clipping norms.
>
> > **Explicitly constructing initialization with 0 gradient norm**
>
> Unfortunately, even if the other gradients’ norms are exactly 0 on iteration 0, it will no longer be 0 for future iterations as the model weights become “corrupted” by the noise addition step [27]. Therefore, to the best of our knowledge, it is not possible to have an initialization that results in the other gradients’ norms being 0 throughout training. Nevertheless, one setting we did consider (but did not include in the paper) is the worst-case neighboring datasets setting, where  $D = \emptyset, D’ = \\{(x, y)\\}$, which is numerically equivalent to what you suggest (the gradient norm of the empty dataset will be 0). While the audits were significantly tighter in this setting ($\varepsilon_{emp} \approx 9$ at theoretical $\varepsilon = 10.0$ for both MNIST and CIFAR-10), we did not include it in our final paper since this setting has limited real-world value.

---

> > ### Author Response · Authors · 2024-08-13
> >
> > We once again thank reviewer qSii for their time and constructive feedback. As the discussion period draws to a close, we hope our rebuttal has sufficiently addressed the reviewer's concerns and would be grateful to hear the reviewer's thoughts on our comments and clarifications.

---

### Author Rebuttal · Authors · 2024-08-02

We thank the reviewers for their insightful feedback and suggestions. In this message, we clarify points mentioned by multiple reviewers. We also address each reviewer’s concerns separately in individual comments.

**1. Novelty of worst-case initial model parameters (qSii, JVDo)**

As also noted by Reviewers ncD1 and uyKT, one of the main contributions of our paper is the novel approach of crafting worst-case initial model parameters, which results in a strong improvement over prior work. While our method is built on similar intuitions (e.g., reducing the contributions of other samples), it is significantly different from prior work [18, 27] and much more effective.

More precisely, while [18] does look at initialization, it focuses on tweaking the randomness of initialization rather than specifically crafting worst-case initial model parameters, thus resulting in looser audits. Furthermore, [27] crafts an adversarial worst-case dataset but sets the learning rate to 0, which destroys the model’s utility.

Designing an effective adversarial strategy that provides tight audits in the black-box setting without destroying utility has remained an open problem. Our work overcomes this challenge through a novel strategy involving crafting worst-case initial model parameters. As also detailed in our comments to each review, in the revised version of the paper, we will clarify our motivation and contribution more clearly while toning down certain claims.

**2. Discrepancies in Figure 5 (uyKT, JVDo)**

The last-layer-only fine-tuning setting is a common setting studied both when auditing DP-SGD and for training private models with DP-SGD. This motivates us to verify whether our method offers improvements in auditing for this setting as well.

As pointed out by Reviewers uyKT and JVDo, we acknowledge that the improvements in this setting are fairly limited.

We also agree that some discrepancies in Figure 5 were not well explained in the text. We will provide a brief explanation here and will explain this clearly in the revised manuscript. Specifically, the mean of the $\varepsilon_{emp}$ for theoretical $\varepsilon = 1.0$ appears to be tighter for the average-case initial model parameters setting rather than the worst-case initial model parameters. However, both values are still within the confidence interval of each other and therefore, we do not consider this to mean that average-case initialization performs better in this setting. Rather, this was an artifact of auditing using a small number of models (200). In fact, in preliminary testing, when the number of models is increased to 1,000, we find that $\varepsilon_{emp} = 0.35 \pm 0.38$ for average-case and $0.51 \pm 0.52$ for worst-case initial model parameters thus showing that there is in fact no discrepancy in Figure 5. We will explain this clearly in the text and will increase the number of models used to audit just for this setting given that we are able to have access to more compute within the timeframe.

**3. Details on worst-case initial model parameters (ncD1, JVDo)**

We thank the Reviewers for highlighting that some details (e.g., optimizer) are missing.

For MNIST, we pre-train the model on half of the dataset using non-private SGD for five epochs with batch size 32 and a learning rate of 0.01.

For CIFAR-10, we pre-train the model first on the CIFAR-100 dataset (there is a typo in the text “ImageNet-32” => “CIFAR-100”, which we will fix) for 300 epochs with batch size 128 and learning rate 0.1 using non-private SGD. We further fine-tune the model on half of the CIFAR-10 dataset using non-private SGD for 100 epochs with batch size 256 and learning rate 0.1.

We will make the details more explicit by moving the text from Appendix A to Section 4.

---

### Decision · Program_Chairs · 2024-09-25

**Decision:**

Accept (poster)

**Comment:**

The reviews are generally positive, and the overall sentiment further improved slightly following the author rebuttal and discussion. There was a debate as to whether the limited technical novelty (running existing auditing techniques on pre-trained models) should be a barrier for acceptance. In the end, the consensus was that the work has some value by evaluating a simple idea which leads to significant practical gains for privacy auditing. Therefore, the paper is accepted.

Important note: the authors have promised to tone down their claims of “near tight” audits to specify in which settings the audits are appreciably tighter than prior work, as requested by some reviewers. Additionally, they have been asked to clearly discuss the limitation of auditing only DP-GD and not DP-SGD (i.e., without subsampling). I therefore ask the authors to incorporate these changes in the final version.